# Endometrioma ethanol sclerotherapy could increase IVF live birth rate in women with moderate-severe endometriosis

Laura Miquel[1]☯*, Lise Preaubert[1]☯, Audrey Gnisci[1], Noémie Resseguier[2], Audrey Pivano[1], Jeanne Perrin[1,3], Blandine Courbiere[1,3]

**1** Department of Obstetrics Gynecology and Reproductive Medicine, Pôle femmes parents enfants, IVF Unit/CECOS, AP-HM La Conception University Hospital, Marseille, France, **2** Research Unit EA 3279, Department of Public Health, Aix-Marseille University, Marseille, France, **3** Aix Marseille Univ, Avignon Université, CNRS, IRD, IMBE, Marseille, France

☯ These authors contributed equally to this work.
* lo.miquel@yahoo.fr

**Data Availability Statement:** All primary files are available from the dryad database (https://doi.org/10.5061/dryad.vx0k6djpq).

## Abstract

### Objective

To examine the impact of ethanol sclerotherapy (EST) for endometrioma on in vitro fertilization (IVF) cumulative live birth rates (CLBR) in women with moderate-severe endometriosis.

### Methods

This retrospective cohort study included women with moderate-severe endometriosis (revised American Fertility Society stage III-IV) and endometrioma who underwent IVF with the ultra-long agonist protocol. We compared two groups: women undergoing EST for endometrioma before IVF (EST group), and women whose endometrioma was left in situ during IVF (No-EST group). The primary outcome was the CLBR per IVF cycle, including fresh and frozen embryo transfers. The secondary endpoints included the complication rate, number of mature oocytes retrieved, clinical pregnancy rate and pregnancy loss rate.

### Results

Seventy-four women were included in the study, with 37 in the EST group and 37 in the No-EST group, representing 67 and 69 IVF cycles, respectively. The population and cycle characteristics were comparable between the two groups, especially the ovarian response to stimulation. The CLBR was significantly increased in the EST group compared to the No-EST group (31.3% vs. 14.5%, p = 0.03). The clinical and biochemical pregnancy rates were significantly increased in the EST group (37.3% vs. 15.9%, p = 0.01 and 43.3% vs. 23.2%, p = 0.01, respectively). Multivariate analysis revealed a significantly increased chance of live birth in women exposed to EST before IVF with an adjusted OR of 2.68 (95% confidence interval, CI: 1.13–6.36, p = 0.02). In the EST group, we reported one major complication Clavien and Dindo classification grade III, complication involving an ovarian abscess that required a laparoscopic drainage.

**Funding:** The author(s) received no specific funding for this work.

**Competing interests:** The authors have declared that no competing interests exist.

## Conclusions

EST is an interesting technique to improve IVF success rates in women with moderate-severe endometriosis. EST could be discussed before IVF in infertile women.

## Introduction

Endometriosis affects 20% to 40% of women consulting for fertility disorders [1]. In particular, women with moderate-severe endometriosis (revised American Fertility Society (rAFS) stage III-IV) have lower pregnancy rates in IVF compared to infertile women with mild to moderate endometriosis (r-AFS stage I-II) or women with tubal factor [2, 3]. Endometriomas are detected in up to 44% of women with endometriosis and are associated with pelvic pain, deep lesions and disease severity [4]. Surgical treatment remains the recommended strategy in symptomatic women and is associated with significantly increased spontaneous pregnancy rates in infertile women [5–7]. However, evidence suggests that the ovarian reserve is injured following surgical excision of endometriomas [8], especially in the case of multiple interventions. The ovarian reserve is established during fetal life with a stock of reserve follicles that decreases from birth to menopause. Any alteration of the ovarian reserve is irreversible and may lead to premature ovarian failure [9]. Conventional endometriosis laparoscopic surgery with divergent cystectomy traction may compromise the ovarian reserve by the excision of the surrounding ovarian cortex. Roustan et al. demonstrated that women with a diminished ovarian reserve after endometrioma cystectomy had significantly lower live birth rates compared with a control group with an idiopathic diminished ovarian reserve [10]. For this reason, routine excision of endometriomas before IVF is currently not recommended unless the endometrioma is painful or hampers oocyte retrieval [11, 12]. Left in situ, endometriomas are associated with significantly lower ovarian response to controlled ovarian stimulation (COS) and incomplete follicular aspiration as demonstrated by reduced numbers of follicles and oocytes retrieved [13, 14]. Moreover, the cyst often has to be transfixed to obtain all the follicles, which involves a risk of contaminating the follicular fluid, potentially affecting live birth rates and contributing to an increased risk of post-retrieval infection [15]. Various agents have been used for intra-cystic instillation with the aim to induce the direct destruction of endometriosis present in the cyst wall: tetracycline [16], methotrexate [17, 18], and ethanol [19–22]. Ethanol sclerotherapy for endometrioma (EST) was first described in 1988 in Japan [23] and later standardized by Yasbeck et al. [22]. Recurrence rates after EST range from 12.9 to 20% and are comparable to those found after laparoscopic cystectomy [24–26]. EST is offered to women with recurrent endometriomas [27] to preserve the ovarian reserve, avoiding another surgical excision [28]. Yazbeck et al. reported increased IVF pregnancy rates after EST compared to women undergoing a second laparoscopic cystectomy [22]. There are few studies, and the results are contradictory. Some studies demonstrated a benefit [20, 22, 29–31], whereas others did not show any significant improvement. Currently, the role of EST in the management of endometriomas is not well defined as demonstrated by the meta-analysis of Cohen et al. [19].

To the best of our knowledge, no studies have investigated IVF cumulative live birth rates (CLBR) after EST. Our aim was to examine the impact of EST for endometrioma on IVF CLBR in women with moderate-severe endometriosis. In particular, we compared women with ethanol sclerotherapy before IVF with women in whom endometrioma(s) was left in situ during IVF. Our main secondary endpoints included the complication rate, the response to

ovarian stimulation (number of mature oocytes retrieved), clinical pregnancy and pregnancy loss rates.

## Materials and methods

### Study population

This single-center retrospective cohort study at the Department of Reproductive Medicine of a University Hospital included infertile women aged 18 to 43 years old presenting with rAFS stage III or IV endometriosis with at least one endometrioma and an indication for IVF between May 2013 and May 2017. All women were followed-up until May 2020.

Women undergoing surgery immediately before IVF for the treatment of endometriosis were not included. However, we did not exclude women with a history of previous laparoscopy and cystectomy if they presented with an endometrioma recurrence at the time of IVF. Endometriomas were defined as endometriotic ovarian cysts between 25 to 65 mm in diameter [32]. The diagnostic was made by trained gynecologists on transvaginal ultrasound and then confirmed by pelvic magnetic resonance imaging (MRI) for all women by expert radiologists specialized in endometriosis imaging. Although the rAFS classification is defined by a laparoscopic evaluation, it is widely admitted that the MRI evaluation is effective for diagnosing moderate-severe endometriosis when performed by experienced radiologists [33–36].

Our cohort included two groups: women having had an EST before IVF (EST group) and women whose endometrioma was left in situ during IVF (No-EST group). Given the lack of consensus concerning EST indications before IVF, some physicians of our IVF unit systematically perform EST, whereas the others do not. The EST group was compared to a group of women who did not have EST before IVF during the same period. Women were matched (1:1) by age, body mass index (BMI) and smoking status. Individual characteristics considered included age, BMI, smoking status, ovarian reserve (antral follicle count on day 3 and Anti-Müllerian Hormone (AMH) dosage), endometriosis severity evaluated by MRI or laparoscopy, duration of infertility, association of a male factor, number of previous IVF cycles, and history of laparoscopy.

The Aix Marseille University's ethics committee approved this study (2018-24-01-008). All women provided written informed consent.

### Ethanol sclerotherapy

We followed a standardized sclerotherapy procedure as previously described by Yasbeck et al., which consists of an ultrasound-guided transvaginal puncture of one or more endometriomas [32]. The intervention was performed on an outpatient basis in the lithotomic position by a trained gynecologist under general anesthesia and transvaginal ultrasound control (Hitachi EUB-5500). Antibiotic prophylaxis with 2g Cefazolin was systematically administered. All women had a negative vaginal sample one week before the procedure. Position and size of the endometrioma were initially recorded. Then, the endometrioma was transvaginally punctured under ultrasound guidance using a single lumen 17 Gx250 mm needle. The cyst content was completely aspirated, and its volume was measured in ml. Part of the liquid was sent to cytology for analysis. The cyst was flushed with saline solution until obtaining a clear liquid. Then, 96% ethanol was injected at 60% of the initial volume of the endometrioma to not over-distend and to avoid any rupture of the cyst and ethanol diffusion into the pelvis. The maximum volume of 96% ethanol for intracystic injection was 60 ml. Ethanol remained in the cyst for 10 minutes and was completely reaspirated. The EST procedure was considered successful if the remaining endometrioma measured less than 20 mm after three months [32]. EST complications were classified according to the Clavien and Dindo surgical classification (Grade I, which

indicates any deviation from the normal postoperative course, to Grade IV, which indicates life-threatening complications) [37].

## IVF procedure

As recommended by the European Society of Human Reproduction and Embryology (ESHRE), the IVF protocol was an ultra-long-agonist protocol consisting of an intra-muscular injection of 3 mg Triptoreline (Decapeptyl, Ipsen, France) at least three months before the EST procedure [11]. Controlled ovarian stimulation was initiated 2 weeks after the EST procedure. The dose and type of gonadotropins were determined on individual basis according to data from previous IVF cycles, age, BMI and ovarian reserve. All women underwent serial transvaginal ultrasounds and hormonal dosages during stimulation, and human chorionic gonadotropin (HCG) was administered subcutaneously (Ovitrelle, Merck Serono, Germany) when three or more leading follicles with a mean diameter greater than 18 mm were visualized. Oocyte retrieval was performed transvaginally 36 h later. Intracytoplasmic sperm injection (ICSI) was performed if necessary according to the sperm characteristics. After IVF or ICSI, one to two embryos were transferred on the second, third or fifth day after retrieval using a soft transfer catheter under ultrasound control. Supernumerary embryos were cryopreserved on days 2, 3 or 5 for subsequent embryo transfer.

## Outcome measures

The primary outcome was the CLBR per IVF cycle, including fresh and frozen embryo transfers (FET). Live birth was defined by a live birth after 22 weeks of gestation according to The International Glossary on Infertility and Fertility Care [38]. Only the first delivery was counted in the analysis if a patient achieved multiple deliveries. Secondary outcomes included peroperative and post-operative complications related to EST, number of mature oocytes retrieved per cycle, fertilization rate, number of diploid embryos per cycle, number of "top" embryos per cycle, number of cryopreserved embryos per cycle, number of embryos transferred per cycle, implantation rate (number of intrauterine gestational sacs divided by the number of transferred embryos), cumulated biochemical pregnancy rate, cumulated clinical pregnancy rates, and pregnancy loss rate. Standard definitions were used as follows: biochemical pregnancy was defined as positive HCG 15 days after embryo transfer, and clinical pregnancy was defined as the visualization of a positive fetal heartbeat on ultrasound at 7 weeks.

## Statistical analysis

Statistical analyses were performed using R software version 3.4.1. Continuous variables are presented as means ± standard deviations, and categorical variables are presented as numbers (percentages). Women exposed to EST were selected and matched in a 1:1 fashion to women who were not exposed to EST. Matching criteria included age (+/- 2 years), BMI (+/- 3 kgs/m$^2$), and smoking status. For univariate analysis, we used Student, Chi2, and Fisher exact tests, as appropriate. Comparisons between cycles and their issues between groups were performed using classical statistical tests (Student t test for quantitative parameters and chi-squared test for categorical parameters) that were adapted for clustered data, which allows consideration of the correlation that might exist between cycle characteristics within a given woman. All tests were bilateral, and a P-value <0.05 was considered statistically significant. Assumption of normality of the distribution for continuous variables was assessed graphically (histograms and quantile-quantile plots). Median and [1st quartile- 3rd quartile] were presented when the assumption of normality was not verified. A multivariate logistic regression model was used to assess the adjusted odds ratio (aOR) of live birth. The model included classical variables

known to influence the chances of success (live birth): age, smoking status, and number of previous IVF cycles. A post-hoc power analysis was performed, and a power of 68% was identified [39].

## Results

Among 2148 women who underwent IVF-ICSI in our center between May 2013 and May 2017, 381 had endometriosis, and 294 had moderate-severe endometriosis. Among those 294 women, 191 women had ovarian endometriosis. Among those 191 women, 158 women still had ovarian endometriosis immediately before IVF. Among these 158 women, 123 women had endometrioma measuring 25 to 65 mm in diameter. Among these 123 women, 6 women underwent a laparoscopic management of the endometrioma immediately before IVF, 77 women did not undergo any treatment and 40 woman underwent sclerotherapy. Among these 117 woman eligible for the study, 99 give their consent the study. Among those 99 woman selected for the study, 37 women had undergone EST and were matched in a 1:1 fashion to women who were not exposed to EST. The median follow-up among patients who obtained a pregnancy was 13.9 ± 7.7 months for the EST group and 9.2 ± 8.5 months for the No EST group. The median follow-up among patients who did not obtained a pregnancy was 51.1 ± 10.1 months for the EST group and 72.0 ± 18.9 months for the No EST group. There was no loss to follow-up.

These 74 women represented a total of 67 IVF cycles in the EST group and 69 IVF cycles in the No-EST group. In the EST group, 43.2% of women had a history of previous laparoscopy for endometrioma treatment vs. 37.8% in the No-EST group (p = 0.64). Population characteristics were comparable between the two groups (Table 1).

The IVF cycle characteristics were not significantly different between the EST group and the No-EST group (Table 2).

The CLBR was significantly increased in the EST group compared to the No-EST group (31.3% vs. 14.5%, p = 0.03). Multivariate analysis showed a significantly increased the chance of live birth in women exposed to EST before IVF. The adjusted OR of live birth was 2.68 (95% confidence interval, CI: 1.13–6.36, p = 0.02).

Clinical and biochemical pregnancy rates were significantly increased in the EST group (37.3% vs. 15.9%, p = 0.01 and 43.3% vs. 23.2%, p = 0.01, respectively). The differences in pregnancy loss rate and implantation rate were not significant between the two groups (respectively 26.7% vs. 37.5%, p = 0.45 and 21.0% vs. 10.7%, p = 0.06).

In the EST group, 8 women underwent EST for bilateral endometriomas, and one iterative procedure was performed in one woman for endometrioma recurrence 6 months later. The mean diameter of the endometriomas eligible at EST in the EST group was 37.3 ± 9.4 mm. In the No-EST group, the mean diameter of the endometriomas eligible at EST was 34.6 ± 11.4 mm. The difference was not significant between the two groups (p = 0.27). EST was successful in 87% of the procedures (40/46). The mean reduction of endometriomas after EST was 28.2% ± 10.6% with a mean final diameter of 9.1 ± 9.5 mm (p <0.01).

In 6 procedures (13.0%), a persistent endometrioma (>20 mm) was observed on ultrasound during IVF (6/46), but its size was at least 5 mm smaller than the original endometrioma. In these cases, the mean reduction of the endometrioma was 27.8% ± 18.5% with a mean final diameter of 29.4 ± 4.9 mm.

Three women presented an immediate post-EST complication. Specifically, two women presented with fever and pelvic pain, amongst whom one resolved after oral antibiotic therapy (Clavien and Dindo classification Grade II) and one required a laparoscopic drainage of an ovarian abscess (Clavien and Dindo classification grade III). A woman presented with

**Table 1. Characteristics of infertile women with moderate-severe endometriosis: Ethanol sclerotherapy (EST) (n = 37) vs. No-EST group (n = 37).**

|  | EST group (n = 37) | No-EST group (n = 37) | p-value |
|---|---|---|---|
| Age (years) | 31.5 ± 4.5 | 33.0 ± 3.8 | 0.13 |
| Tobacco smoker | 8 (21.6) | 6 (16.2) | 0.55 |
| BMI (kg/m$^2$) | 22.8 ± 4.7 | 24.6 ± 6.2 | 0.16 |
| Day 3 FSH (IU/L) | 6.5 ± 2.1 | 7.7 ± 3.4 | 0.08 |
| Day 3 estradiol (pg/mL) | 44.0 ± 21.2 | 44.2 ± 18.6 | 0.97 |
| AMH (ng/ml) | 3.2 ± 2.6 | 3.1 ± 2.2 | 0.90 |
| AFC | 11.1 ± 6.3 | 12.4 ± 5.6 | 0.36 |
| Duration of infertility (years) | 4.4 ± 3.1 | 4.3 ± 2.4 | 0.82 |
| Male factor associated | 12 (32.4) | 10 (27.0) | 0.61 |
| Number of previous IVF cycles | 0.4 ± 0.9 | 0.2 ± 0.4 | 0.11 |
| History of operative laparoscopy for endometriosis global treatment | 14 (37.8) | 13 (35.1) | 0.81 |
| Extra ovarian endometriosis | 26 (70.3) | 25 (67.6) | 0.80 |
| Number of endometrioma | 2.1 ± 1.3 | 2.1 ± 1.3 | 0.86 |
| Bilateral endometriomas | 22 (59.5) | 17 (45.9) | 0.24 |
| Median endometriomas size | 33 [22–39] | 27 [25–33] | 0.33 |
| **Type of extra ovarian endometriosis** |  |  |  |
| Rectovaginal space or vaginal | 11 (29.7) | 15 (40.5) | 0.33 |
| Uterosacral and cardinal ligaments | 21 (56.8) | 15 (40.5) | 0.16 |
| Rectosigmoid | 13 (35.1) | 11 (29.7) | 0.62 |
| Bladder or ureter | 2 (5.4) | 4 (10.8) | 0.67 |
| Other bowel involvement and extragenital localizations | 2 (5.4) | 2 (5.4) | 1.00 |
| Adenomyosis | 5 (13.5) | 3 (8.1) | 0.71 |

Data are expressed as the mean ± SD or *n* (%) if not specified. Median are expressed with [1st quartile- 3rd quartile]. BMI = body mass index. AFC = antral follicle count on Day 3; AMH = anti-Müllerian hormone.

intravascular alcohol diffusion (Clavien and Dindo classification grade I) with a positive blood alcohol concentration without other consequences (0.22 g/L) [37]. No malignant cells were found in the cytological analyses.

Concerning the IVF procedure, during the follow-up after oocyte retrieval, 2 women in the EST group (2.9%) presented with post-retrieval fever and pelvic pain, amongst whom one resolved after antibiotic therapy (Clavien and Dindo classification Grade II) and one required transvaginal drainage of a tubal abscess (Clavien and Dindo classification grade III). In the No-EST group, 5 post-retrieval complications were recorded (7.2%). Five women presenting with fever and pelvic pain were successfully treated by antibiotics (Clavien and Dindo classification Grade II).

## Discussion

In our study, EST before IVF in women with moderate-severe endometriosis was associated with significantly increased CLBR compared with IVF performed with endometrioma left in situ. These results are clinically important given that managing infertile women with an endometrioma before IVF remains a matter of debate [11].

Compared to controls, women with endometriomas have a lower ovarian reserve and ovarian response to COS as demonstrated by the reduced number of retrieved oocytes and mature oocytes [40, 41]. The impact of IVF on an endometriotic cyst seems to be mild. In their systematic review, Somigliana et al. reported that IVF does not worsen endometriosis-related pain symptoms or increase the risk of endometriosis recurrence [42].

**Table 2. Cycle characteristics and IVF outcomes in women with moderate-severe endometriosis: Ethanol sclerotherapy (EST) vs. No-EST group.**

|  | EST group (n = 67 cycles) | No-EST group (n = 69 cycles) | p-value |
|---|---|---|---|
| Total gonadotropin dose (IU) | 3316.4 ± 1374.8 | 2871.1 ± 1082.6 | 0.10 |
| E2 on HCG day (pg/ml) | 2250.5 ± 1295.9 | 2613.6 ± 1534.7 | 0.25 |
| Stimulation days | 11.0 ± 2.0 | 10.7 ± 2.2 | 0.43 |
| No. of mature oocytes | 5.5 ± 3.3 | 5.8 ± 3.8 | 0.71 |
| Fertilization rate (%) | 62.3 ± 30.3 | 58.2 ± 27.7 | 0.46 |
| No. of diploid embryos | 3.4 ± 2.4 | 3.4 ± 3.0 | 0.97 |
| No. of "Top" embryos | 0.3 ± 0.8 | 0.3 ± 0.6 | 0.68 |
| No. of cryopreserved embryos | 0.7 ± 1.3 | 0.4 ± 0.9 | 0.12 |
| No. of embryos transferred | 1.9 ± 1.4 | 1.7 ± 1.0 | 0.31 |
| **Total number of embryos transfers** | **(n = 81)** | **(n = 70)** | **p-value** |
| No. of frozen embryos transfers | 26 (28.0) | 12 (14.8) | 0.08 |
| No. of "Top" embryos per transfer | 0.2 ± 0.5 | 0.1 ± 0.4 | 0.24 |
| Implantation rate | 21.0 ± 37.8 | 10.7 ± 28.1 | 0.06 |
| **Cumulative pregnancy outcomes per cycle** | **(n = 67 cycles)** | **(n = 69 cycles)** | **p-value** |
| Live birth rate | 21 (31.3) | 10 (14.5) | **0.03** |
| Clinical pregnancy rate | 25 (37.3) | 11 (15.9) | **0.01** |
| Biochemical pregnancy rate | 29 (43.3) | 16 (23.2) | **0.01** |
| Pregnancy loss rate | 8 (26.7) | 6 (37.5) | 0.45 |

Data are expressed as the mean ± SD or *n* (%). E2 = estradiol. HCG = human chorionic gonadotropin.

Our results concerning the increased success rates of IVF are consistent with those from previous studies. In the prospective comparative study of women with recurrent endometriomas by Yazbeck et al., IVF pregnancy rates after IVF was increased after EST compared to surgical excision [22]. In the retrospective comparative study of Guo et al., endometriomas aspiration (without sclerotherapy) exhibited improved ovarian response, embryo quality, implantation rates and IVF clinical pregnancy rates compared to conservative management [43]. However, aspiration of endometrioma is not recommended because it is associated with a very high rate of recurrence (28.9%-91.5%) and a risk of complications, such as pelvic infections [44]. In a retrospective study of 101 women, Lee et al. compared EST before IVF to surgical excision or to conservative management [30]. EST and conservative management were associated with improved ovarian reserve and ovarian response to COS compared to surgical excision. However, LBR was comparable in the three groups. Nevertheless, Lee et al. used 20% ethanol for their EST procedure (vs. 96% pure sterile ethanol in our study) and they included women up to one year between EST and IVF, suggesting a possible decrease in the EST benefit with time.

Endometriosis is a chronic, inflammatory, estrogen-dependent disease, in which endometrial stromal cells acquire the capacity to proliferate, migrate outside the uterine cavity, and invade adjacent tissues. One of the main hypotheses is based on retrograde menstruation [34]. However, the mechanisms underlying the endometrial tissue grafting essentially remain unknown. Recent studies support a multifactorial origin, combining anatomical, hormonal, immunological, genetic, epigenetic, and environmental factors [45]. Considering the molecular level, it appears that ion channels, such as cystic fibrosis transmembrane conductance regulator, aquaporins, and chloride channel-3 dysregulation, are potentially involved in the physiopathology of endometriosis [46]. Several hypotheses could explain increased IVF CLBR after EST. Decreasing the size of the endometrioma could reduce the compression of the

ovarian cortex and enhance the vascular flow [25]. This mechanism may explain the significant increase in antral follicular count [47] as observed by some authors [19, 22]. We did not find any significant difference in AFC or ovarian response between our EST and No-EST group possibly due to the moderate size of endometrioma(s) included in our study. The increase in CLBR does not appear to be linked to a better ovarian response to COS or an increase in the number or the quality of the embryos. Our study groups were comparable in terms of number of embryos transfers, embryo quality and number of embryos transferred. In association with an ultra-long agonist protocol, EST could increase IVF LBR by decreasing pelvic and intra-ovarian inflammation that is associated with moderate-severe endometriosis, thereby improving the oocyte quality and implantation [34]. The low implantation rate observed in the No-EST group with an in situ endometrioma could support this hypothesis. We hypothesize that EST with an ultra-long agonist protocol might be associated with a better implantation rate.

EST may also act via a direct mechanism on the ovary. Indeed, the impact of ethanol sclerotherapy on the adjacent tissue remains unclear. The mechanism of action could combine cytotoxic damage, cell dehydration, coagulation and thrombosis [48]. Wang *et al.* identified differentially expressed genes and potential serum biomarkers in women with endometrioma before and after EST [49]. They reported that the expressions of interleukin-6 (IL6), *CD36*, *JUNB*, *B4GALT1*, *HES1*, and *NR4A1* gene, which are strongly associated with the pathogenesis of ovarian endometrioma, was significantly down regulated after EST. The mechanisms of EST remain unclear and should be further investigated.

On the other hand, in a small randomized clinical trial of women with endometriomas recurrence after laparoscopy, Aflatoonian et al. failed to show any difference in clinical pregnancy rates between women with EST and women with conservative management, possibly due to a lack of power [25]. In a retrospective study of 23 women, Suganuma et al. did not observe any significant differences between EST and conservative management regarding the number of retrieved oocytes, mature oocytes, or pregnancy rates [31]. The EST technique and the time between EST and IVF were not described.

The meta-analysis of Cohen et al. included seven studies on EST [19]. Only one was a randomized controlled trial including 40 woman [25]. The heterogeneity of sclerotherapy protocols and outcome measures make it difficult to draw conclusions about the impact of EST on IVF outcomes. The duration of ethanol instillation inside the cyst would be of major relevance given that EST is more efficient if the instillation lasts 10 minutes or more [19].

Our study is the first to compare IVF CLBR in women undergoing EST with an ultra-long agonist IVF protocol vs. ultra-long agonist IVF protocol alone. We were able to access the CLBR without loss to follow-up, whereas previous studies have mainly focused on ovarian response to COS and clinical pregnancy rate. Evaluating CLBR should be the main endpoint for fertility studies, especially in women with endometriosis, as they are supposedly at an increased risk for pregnancy loss [50]. Another strength of our study is the significant number of EST procedures described given that EST is not a routine procedure in many centers and only small numbers are available in the literature. Moreover, we examined the medical records thoroughly and extracted a large number of variables from the fertility and surgery files, which allowed us to analyze the essential information in detail.

In our study, we reported an EST complication rate of 6.5% with one grade III complication according to the Clavien and Dindo classification. The different types and frequency of complications observed are consistent with the literature (postoperative fever, intracystic abscess, alcohol intoxication) [19]. Of note, in the No-EST group, several cases of post-retrieval infections were recorded (7.2%). Endometrioma infection has been described as a classical complication of oocyte retrieval, but its exact incidence is not known and could be underestimated [15]. The small number of reported cases of complications could be due to publication bias. To

prevent any additional risk of infection, a strict sterile technique should systematically be used when performing EST, and most authors also recommend antibiotic prophylaxis.

Despite bias induced by a retrospective survey, including the possibility of confounding factors, our results suggest that EST associated with a long agonist protocol could be an interesting technique to improve IVF success rates in women with moderate-severe endometriosis. To ensure the comparability of our two groups, we performed a matching procedure based on age, BMI and smoking status, which are potential confounding factors.

We limited our analysis to small to medium endometriomas sizes (25–65 mm of diameter); therefore, our results cannot be extrapolated to larger cysts. The EST procedure was based on Yasbeck's protocol and Cohen's meta-analysis [19, 22, 32], which both describe the sclerotherapy of cysts up to 65 mm. However, in our experience, we have gradually increased the size of endometriomas eligible for sclerotherapy from 65 to 100 mm in diameter. A recent study also reports the successful sclerotherapy of endometriomas up to 100 mm [51]. It would be interesting to study IVF live birth rates after EST of cysts measuring 65 to 100 mm. Moreover, the management of endometriosis should be multidisciplinary, reproducible and standardized to improve the quality of care [52]. The development of reference centers dedicated to the care of women with endometriosis is potentially useful to achieve this goal.

EST offers several advantages compared to the laparoscopic approach. EST is a rapid outpatient procedure that requires minimal equipment at a low cost. Importantly, comparing EST with cystectomy, EST preserves the ovarian reserve [22, 30, 53] in a population already at risk for decreased ovarian reserve, especially in the case of recurrence. EST could be discussed before IVF as first-line therapy in infertile women, especially among those with preexisting diminished ovarian reserve. However, the cost-effectiveness of EST should be further investigated, and large randomized studies are needed to better evaluate the CLBR and safety of EST before IVF.

## Supporting information

**S1 Data.**
(XLSX)

## Author Contributions

**Conceptualization:** Audrey Gnisci, Blandine Courbiere.

**Formal analysis:** Noémie Resseguier.

**Investigation:** Audrey Pivano, Jeanne Perrin.

**Methodology:** Noémie Resseguier.

**Resources:** Audrey Pivano.

**Supervision:** Audrey Pivano, Blandine Courbiere.

**Validation:** Blandine Courbiere.

**Writing – original draft:** Laura Miquel, Lise Preaubert.

**Writing – review & editing:** Blandine Courbiere.

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
