## [Decision Letter · Decision Letter 0]

16 Jul 2020

PONE-D-20-17318

Endometrioma ethanol sclerotherapy could increase IVF live birth rate in women with severe endometriosis

PLOS ONE

Dear Dr. MIQUEL,

Thank you for submitting your manuscript to PLOS ONE. After careful consideration, we feel that it has merit but does not fully meet PLOS ONE’s publication criteria as it currently stands. Therefore, we invite you to submit a revised version of the manuscript that addresses the points raised during the review process.

The manuscript and the reviewers’ comments were carefully evaluated. The manuscript was appreciated by Reviewers. Nevertheless, they highlighted different points of concern in the manuscript that need to be revised before to be considered for publication. Suggested revisions are in detail reported in the Reviewers’ comments.

We look forward to receiving your revised manuscript.

Kind regards,

Simone Garzon

Academic Editor

PLOS ONE

Journal Requirements:

Reviewers' comments:

Reviewer's Responses to Questions

**Comments to the Author**

1. Is the manuscript technically sound, and do the data support the conclusions?

Reviewer #1: Yes

Reviewer #2: Partly

Reviewer #3: Yes

Reviewer #4: Partly

2. Has the statistical analysis been performed appropriately and rigorously? 

Reviewer #1: Yes

Reviewer #2: I Don't Know

Reviewer #3: No

Reviewer #4: No

3. Have the authors made all data underlying the findings in their manuscript fully available?

Reviewer #1: Yes

Reviewer #2: Yes

Reviewer #3: Yes

Reviewer #4: Yes

4. Is the manuscript presented in an intelligible fashion and written in standard English?

Reviewer #1: Yes

Reviewer #2: Yes

Reviewer #3: Yes

Reviewer #4: No

5. Review Comments to the Author

Reviewer #1: The authors described the efficacy of endometrioma ethanol sclerotherapy (EST) before IVF. They concluded EST improved IVF success rates in women with severe endometriosis.

The manuscript is well written and novel information is contained. I recommend this article for publication after certain revision.

Major point

1. Add the criteria of EST patient selection. In 294 severe endometriosis women, how the 37 patients decided to be treated with EST? Could the authors mention isn’t there any selection bias on EST patient selection?

Minor point

1. As an indicator of ovarian reserve, add the basal (usually measured on day 3) serum FSH and estradiol and inhibin B value to Table1.

2. How many patients were involved in the bilateral cyst in each group?

Reviewer #2: This is an interesting work on ethanol sclerotherapy that favours IVF in endometriosis, although there is an important selection bias in the cohorts since it depends on the physician. I have some specific comments on the following parts that I describe below

Specific Comments:

Abstract:

1. Methods are not clear: Is the study prospective or retrospective? The cohort groups were not defined.

2. Were all the patients with any type of endometriomas included? Or only when extraovarian endometriosis was associated. The definition of severe endometriosis is unclear.

3. Did the non-EST group receive any specific treatment for endometriosis?

4. How many cycles were included in each group?

5. All secondary endpoints results were not described in the abstract, please select the most relevant.

Introduction

1. The secondary endpoints described in the abstract were not included in the purpose of the main article.

Material and Methods

1. The size of the endometrioma less than 5 cm, especially those of 25 mm, can behave significantly differently from that of size greater than 5 cm

2. If inclusion in each group depends on the physician, why “Women were matched (1:1) by age, Body Mass Index (BMI) and smoking status”

3. ICSI initials must be defined

4. Why did you choose only up to 6 cm of endometrioma size?

5. Perhaps a multivariate analysis of the predictors of pregnancy should improve your work.

Results

1. Why are there only 74 patients included in the study of 294 severe endometriosis?

2. Median endometrioma size in each group should be available, maybe in table 1 better than mean (unless this variable follows the normal curve)

3. Are there any bilateral cases in the no-EST group?

4. It is surprising that 3 cases of EST have a complication of infection after ethanol instillation and prophylactic antibiotic, because other studies and my own experience found none if the procedure was well-done. Is it possible that other technical problems were related to this complication?

5. Were the IVF cycles consecutive or for how long were the patients included in the study?

6. In the EST group if there were 4 pregnancy lost, the live birth rate should be 20…or maybe there are some patients in both groups…can you explain it?

Discussion

1. The weak and strong points of this work should be more highlighted in the discussion

Bibliography

1. Reference 31 and 40 are the same

2. More recent papers maybe useful:

a. Findeklee S, Radosa JC, Hamza A, Haj Hamoud B, Iordache I, Sklavounos P, et al. Treatment algorithm for women with endometriosis in a certified Endometriosis Unit. Minerva Ginecol 2020

b. Garcia-Tejedor A, Martinez-Garcia JM, Candas B, Suarez E, Mañalich L, Gomez M, et al. Ethanol Sclerotherapy versus Laparoscopic Surgery for Endometrioma Treatment. A Prospective, Multicenter, Cohort Pilot Study. J Minim Invasive Gynecol 2020.

Reviewer #3: The manuscript is well written and the topic is relevant.

Introduction: well written

Methods:

- did authors performed a sample size calculation or power analysis?

- was there any drop out/ lost of follow up?

- the results were evaluated by intention to treat? was there any cross between groups?

- patients with adenomyosis were included?

Results:

Could the authors include in table 1/results:

- the type of endometriosis and others sites affected (bowel, vaginal, retrocervical, etc)?

- number of ovaryan endometriomas, unilateral/bylateral, and mean size (before and after EST)?

- previous ovaryan surgery (cystectomy, drainage, laser. ooforectomy, etc)?

- did the authors adjusted the results by age?

- any of included patients were submmited to surgery before the IVF?

Reviewer #4: I read with great interest the Manuscript titled “Endometrioma ethanol sclerotherapy could increase IVF live birth rate in women with severe endometriosis” (PONE-D-20-17318).

The study was approved by the ix Marseille University's ethics committee (2018-24-01-008), and all women gave a written informed consent. The study was aimed to examine the impact of ethanol sclerotherapy for endometriomas (EST) on IVF cumulative live birth rates (CLBR) in women with severe endometriosis.

I was particularly pleased to review this paper. In my honest opinion, the topic is interesting enough to attract the readers’ attention. Nevertheless, authors should clarify some point and improve the discussion citing relevant and novel key articles about the topic and discussion limitations of the study that are not evidenced in the discussion:

- All the text needs a language revision by a native English speaker person, in order to improve some typos and grammatical errors.

- I would suggest checking the guidelines for the Authors to conform the manuscript.

- All the abbreviation should be reported in the extended form at the first use both in the abstract and the main text and tables.

- Abstract. Methods are lacking key information, such as the treatment investigated and how and which groups were selected and compared.

- Methods. I would suggest providing the following pieces of information:

a) How the patients were identified

b) The source of patient list (medical records, registry, DRG)

c) Who extracted data?

d) Source of clinical information

- Regarding endometriosis score, were all patients reassessed by RMI before EST/IVF regardless surgery or for some patients was used the score estimated during surgery? In this case the effect of surgery represents a factor to be considered in the comparison of the two groups. In this regard, is the RMI accuracy valid for patients who had already underwent surgery?

- The match of patients should be reported in the statistical analysis description.

- Statistical analysis. Was the assumption of normal distribution for continues variable assessed?

- Results. The matching of the two groups needs to be better described. How many patients with severe endometriosis underwent EST? How many were eligible for the study? How many patients did not undergo EST? How the two groups were matched based on age, BMI, and smoke? In the example, were matched by age based on age +- 1 year? Being possible to match 1:2 or 1:3, did the authors estimated a sample size calculation? Were matched women who did not received EST randomly selected? Or was a propensity score method used?

- Media follow-up among patients who obtained a pregnancy should be reported. How many patients were missed during follow-up? At which time point does the CLBR refer to?

- I would suggest improving the discussion discussing at least briefly about the etiopathogenesis of endometriosis. Refer to (PMID: 32046116; PMID: 31717614).

- I would suggest, at least briefly, to discuss more about the COS in patients with endometriosis and specifically endometriomas (PMID: 31755673).

6. PLOS authors have the option to publish the peer review history of their article (what does this mean?). If published, this will include your full peer review and any attached files.

Reviewer #1: **Yes: **Akemi Koshiba

Reviewer #2: **Yes: **Garcia-Tejedor, Amparo

Reviewer #3: No

Reviewer #4: No

---

## [Author Response · Author response to Decision Letter 0]

2 Sep 2020

Reviewer #1:

The authors described the efficacy of endometrioma ethanol sclerotherapy (EST) before IVF. They concluded EST improved IVF success rates in women with severe endometriosis. The manuscript is well written and novel information is contained. I recommend this article for publication after certain revision.

Response: We are very grateful for your sound remarks and helpful comments to improve our manuscript. The responses and corrections are listed below, and changes have been highlighted in the manuscript.

Major point :

1. Add the criteria of EST patient selection. In 294 severe endometriosis women, how the 37 patients decided to be treated with EST? Could the authors mention isn’t there any selection bias on EST patient selection?

Response: We thank you for pointing out that our formulation was not clear. We have added the following sentences accordingly:

Line 167: “Among those 294 women, 191 women had ovarian endometriosis. Among those 191 women, 158 women still had ovarian endometriosis immediately before IVF. Among these 158 women, 123 women had endometrioma measuring 25 to 65 mm in diameter. Among these 123 women, 6 women underwent a laparoscopic management of the endometrioma immediately before IVF, 77 women did not undergo any treatment and 40 woman underwent sclerotherapy. Among these 117 woman eligible for the study, 99 give their consent the study. Among those 99 woman selected for the study, 37 women had undergone EST and were matched in a 1:1 fashion to women who were not exposed to EST.”

As explained line 87 because of the lack of consensus concerning EST indications before IVF, some physicians of our IVF unit systematically perform EST, whereas the others do not. There is therefore a selection bias which we have tried to erase through patient matching.

Minor point

1. As an indicator of ovarian reserve, add the basal (usually measured on day 3) serum FSH and estradiol and inhibin B value to Table1.

Response: We added the basal serum FSH and estradiol values to Table 1, as requested. However, in our practice we do not routinely ask for inhibin B dosage, so adding this variable was not possible.

Line 186: 

Day 3 FSH (IU/L) 6.5 ± 2.1 7.7 ± 3.4 0.08

Day 3 estradiol (pg/mL) 44.0 ± 21.2 44.2 ± 18.6 0.97

2. How many patients were involved in the bilateral cyst in each group?

Response: In the EST group a total of 22 women presented with bilateral endometriomas compared to 17 women in the No-EST group (59.5% vs. 45.9%, p = 0.24).

This has been added in table 1.

Line 186:

Bilateral endometriomas 22 (59.5) 17 (45.9) 0.24

Reviewer #2: 

This is an interesting work on ethanol sclerotherapy that favours IVF in endometriosis, although there is an important selection bias in the cohorts since it depends on the physician. I have some specific comments on the following parts that I describe below

Response: We have read your comments with close attention and we thank you very much for your review which clearly helped us to improve our manuscript. The responses and corrections are listed below, and changes have been highlighted in the manuscript.

Specific Comments:

Abstract:

1. Methods are not clear: Is the study prospective or retrospective? The cohort groups were not defined.

Response: We agree with that comment. The abstract has been corrected accordingly:

Line 5: “This retrospective cohort study included women with moderate-severe endometriosis (revised American Fertility Society stage III-IV) and endometrioma(s) who underwent IVF with ultra-long agonist protocol. We compared two groups: women undergoing EST for endometrioma before IVF (EST group), and women whose endometrioma was left in situ during IVF (No-EST group).”

2. Were all the patients with any type of endometriomas included? Or only when extraovarian endometriosis was associated. The definition of severe endometriosis is unclear.

Response: All types of endometriomas were included, as long as they measured 25 to 65 mm in diameter. So some of them were isolated whereas others were associated with extra-ovarian lesions. Moderate-severe endometriosis is comprised of stage III and IV endometriosis according to the revised American Fertility Society (rAFS) classification (American Society for Reproductive Medicine. Revised American Society for Reproductive Medicine classification of endometriosis: 1996. Fertil Steril. 1 mai 1997;67(5):817‑21.) We corrected the title, abstract and text in order to clarify this point, as follows: 

Title : “Endometrioma ethanol sclerotherapy could increase IVF live birth rate in women with moderate-severe endometriosis”

Line 2: “To examine the impact of ethanol sclerotherapy for endometriomas (EST) for endometrioma on In Vitro Fertilization (IVF) cumulative live birth rates (CLBR) in women with moderate-severe endometriosis”

Line 5: “This retrospective cohort study included women with moderate-severe endometriosis (revised American Fertility Society stage III-IV) and endometrioma(s) who underwent IVF with ultra-long agonist protocol.”

Line 25: “EST may be is an interesting technique to improve IVF success rates in women with moderate-severe endometriosis. EST could be discussed before IVF in infertile women.”

Line 27: “In particular, women with moderate-severe endometriosis (revised American Fertility Society (rAFS) stage III-IV) have lower pregnancy rates in IVF, when compared to infertile women with mild to moderate endometriosis (r-AFS stage I-II) or women with tubal factor (2,3)”

Line 63: “Our aim was to examine the impact of EST for endometrioma on IVF CLBR, in women with moderate-severe endometriosis.”

Line 82: “Although the rAFS classification is defined by a laparoscopic evaluation, it is widely admitted that the MRI evaluation is effective for diagnosing moderate-severe endometriosis when performed by experienced radiologists (33–36)”

Line 166: “Among 2148 women who underwent IVF-ICSI in our center between May 2013 and May 2017, 381 had endometriosis, and 294 had moderate-severe endometriosis.”

Line 184: “Table 1: Characteristics of infertile women with moderate-severe endometriosis: ethanol sclerotherapy (EST) (n = 37) vs. No-EST group (n = 37).”

Line 192: “Table 2: Cycle characteristics and IVF outcomes in women with moderate-severe endometriosis: ethanol sclerotherapy (EST) vs. No-EST group.”

Line 231: “In our study, EST before IVF in women with moderate-severe endometriosis was associated with significantly higher increased CLBR compared than with IVF performed with endometrioma left in situ.”

Line 272: “In association with an ultra-long agonist protocol, EST could increase IVF LBR by decreasing pelvic and intra-ovarian inflammation, that is associated with moderate-severe endometriosis, therefore thereby improving the oocyte quality and implantation (34)”

Line 318: “Despite bias induced by a retrospective survey, including the possibility of confounding factors, our results suggest that EST associated to with a long agonist protocol could be an interesting technique to improve IVF success rates in women with moderate-severe endometriosis.”

3. Did the non-EST group receive any specific treatment for endometriosis?

Response: Thank you for raising this point. The No-EST group did not receive any specific treatment for endometriosis before starting the ultra-long agonist protocol. The text has been corrected as follows: 

Line 7: “We compared two groups: women undergoing EST for endometrioma before IVF (EST group), and women whose endometrioma was left in situ during IVF (No-EST group).”

4. How many cycles were included in each group?

Response: we included a total of 67 IVF cycles in the EST group and 69 IVF cycles in the No-EST group. As this does not seem to be as clear as we wanted it to be, we have preferred to modify the text as follows:

Line 13: “A total of 67 IVF cycles were included in the EST group and 69 in the no-EST group. Seventy-four women were included in the study, 37 in the EST group and 37 in the No-EST group, representing 67 and 69 IVF cycles, respectively.”

5. All secondary endpoints results were not described in the abstract, please select the most relevant.

Response: The text has been corrected as follows:

Line 10: “The secondary endpoints included the complication rate, number of mature oocytes retrieved, number of diploid embryos, clinical pregnancy rate and pregnancy loss rate.”

Introduction

1. The secondary endpoints described in the abstract were not included in the purpose of the main article.

Response: Thank you for this remark. The main secondary endpoints have been added to the introduction, as follows: 

Line 66: “Our main secondary endpoints included the complication rate, the response to ovarian stimulation (number of mature oocytes retrieved), clinical pregnancy and pregnancy loss rates.”

Material and Methods

1. The size of the endometrioma less than 5 cm, especially those of 25 mm, can behave significantly differently from that of size greater than 5 cm

Response: We totally agree with this remark concerning pain management. But concerning the management of oocyte retrieval, the cyst, even if it is smaller than 5 cm often has to be transfixed to get all the follicles. This involve a risk of contaminating the follicular fluid, which could affect live birth rates and contribute to a higher risk of post-retrieval infection:

Line 49: “Moreover, the cyst often has to be transfixed to get obtain all the follicles, which involvinges a risk of contaminating the follicular fluid, potentially which could affect affecting live birth rates and contribute to an higher risk of post-retrieval infection (15).”

2. If inclusion in each group depends on the physician, why “Women were matched (1:1) by age, Body Mass Index (BMI) and smoking status”

Response: First, among all IVF patients diagnosed with severe endometriosis, we retrospectively identified all women having had an ethanol sclerotherapy for an endometrioma before IVF during the study period. 

Line 167: “Among those 294 women, 191 women had ovarian endometriosis. Among those 191 women, 158 women still had ovarian endometriosis immediately before IVF. Among these 158 women, 123 women had endometrioma measuring 25 to 65 mm in diameter. Among these 123 women, 6 women underwent a laparoscopic management of the endometrioma immediately before IVF, 77 women did not undergo any treatment and 40 woman underwent sclerotherapy. Among these 117 woman eligible for the study, 99 give their consent the study. Among those 99 woman selected for the study, 37 women had undergone EST and were matched in a 1:1 fashion to women who were not exposed to EST.”

In our practice, some physicians systematically perform EST, whereas the others do not. There is therefore a selection bias which we have tried to erase through patient matching.

Line 90: Women were matched (1:1) by age, body mass index (BMI) and smoking status. Individual characteristics considered included were: age, BMI, smoking status, ovarian reserve (antral follicle count on day 3 and Anti-Müllerian Hormone (AMH) dosage), endometriosis severity evaluated by MRI or laparoscopy, duration of infertility, association of a male factor, number of previous IVF cycles, and history of laparoscopy. 

3. ICSI initials must be defined

Response: This has been corrected and the change highlighted in the manuscript.

Line 128: “Intracytoplasmic sperm injection (ICSI) was carried out performed if necessary, according to the sperm characteristics.”

4. Why did you choose only up to 6 cm of endometrioma size?

Response: The EST procedure was based on Yasbeck's protocol and Cohen’s meta-analysis (Yazbeck C, Koskas M, Cohen Scali S, Kahn V, Luton D, Madelenat P. [How I do... ethanol sclerotherapy for ovarian endometriomas]. Gynécologie Obstétrique Fertil. oct 2012;40(10):620‑2 ; Cohen A, Almog B, Tulandi T. Sclerotherapy in the management of ovarian endometrioma: systematic review and meta-analysis. Fertil Steril. juill 2017;108(1):117-124.e5.), which both describe the sclerotherapy of cysts up to 65 mm.

5. Perhaps a multivariate analysis of the predictors of pregnancy should improve your work.

Response: We agree with this remark. This allows us to considerably increase the quality of our work. We performed a multivariate analysis of the main predictors of live birth: age, smoking status, and number of previous IVF cycles. 

The manuscript has been corrected as follows:

Line 19: " Multivariate analysis revealed a significantly increased chance of live birth in women exposed to EST before IVF with an adjusted OR of 2.68 (95% confidence interval, CI: 1.13–6.36, p = 0.02)."

Line 161: “A multivariate logistic regression model was used to assess the adjusted odds ratio (aOR) of live birth. The model included classical variables known to influence the chances of success (live birth): age, smoking status, and number of previous IVF cycles.”

Line 198: " Multivariate analysis showed a significantly increased the chance of live birth in women exposed to EST before IVF. The adjusted OR of live birth was 2.68 (95% confidence interval, CI: 1.13–6.36 p = 0.02)"

Results

1. Why are there only 74 patients included in the study of 294 severe endometriosis?

Response: We thank you for pointing out that our formulation was not clear. We have added the following sentences accordingly:

Line 167: “Among those 294 women, 191 women had ovarian endometriosis. Among those 191 women, 158 women still had ovarian endometriosis immediately before IVF. Among these 158 women, 123 women had endometrioma measuring 25 to 65 mm in diameter. Among these 123 women, 6 women underwent a laparoscopic management of the endometrioma immediately before IVF, 77 women did not undergo any treatment and 40 woman underwent sclerotherapy. Among these 117 woman eligible for the study, 99 give their consent the study. Among those 99 woman selected for the study, 37 women had undergone EST and were matched in a 1:1 fashion to women who were not exposed to EST.”

2. Median endometrioma size in each group should be available, maybe in table 1 better than mean (unless this variable follows the normal curve)

Response: The endometriomas median of endometriomas eligible for an EST in each group have been added to Table 1:

Line 168:

Median endometriomas size 33 [22-39] 27 [25-33] 0.33

3. Are there any bilateral cases in the no-EST group?

Response: In the EST group, a total of 22 women presented with bilateral endometriomas compared to 17 women in the No-EST group (59.5% vs. 45.9%, p = 0.24).

This has been added in table 1:

Line 168:

Bilateral endometriomas 22 (59.5) 17 (45.9) 0.24

4. It is surprising that 3 cases of EST have a complication of infection after ethanol instillation and prophylactic antibiotic, because other studies and my own experience found none if the procedure was well-done. Is it possible that other technical problems were related to this complication?

Response: We were also surprised to find such a high rate of infection in our serie all the more as we are an IVF centre in a university hospital with many years of experience in transvaginal punctures (Agostini A, De Lapparent T, Collette E, Capelle M, Cravello L, Blanc B. In situ methotrexate injection for treatment of recurrent endometriotic cysts. Eur J Obstet Gynecol Reprod Biol. janv 2007;130(1):129‑31.) We found only cases of infection and one other complication. 

In the first case, the sclerotherapy was performed without any particular complications. It was a bilateral procedure with the puncture of 2 endometriomas of 149cc and 23cc.

During the following 7 days, the woman presented with a slight fever and pelvic pain. Biological examinations were performed in the presence of a biological inflammatory syndrome. Oral antibiotic therapy with Cefixime was started and 3 days later, changed to intravenous antibiotic therapy with Ceftriaxone and Metronidazole. Ultrasound revealed an ovarian mass equivalent to that of the initial endometrioma. In the absence of clinical improvement 3 days later, laparoscopy for surgical drainage was performed. The patient then rapidly recovered.

In the second case, the sclerotherapy was also performed without any particular complication.

Four days later, the women presented with pelvic pain and urinary burning. The pelvic exam found an impaction of the Douglas corresponding to the puncture site of the endometrioma. There was also a biological inflammatory syndrome. The patient was placed on Ciprofloxacin and Metronidazole and then improved.

The third complication was a leakage of ethanol into the pelvis resulting in premature withdrawal of the procedure and a positive blood alcohol level in the patient's postoperative period.

Endometrioma infection has been described as a classical complication of oocyte retrieval, but its exact incidence after sclerotherapy is not known and could be underestimated (Cohen A, Almog B, Tulandi T. Sclerotherapy in the management of ovarian endometrioma: systematic review and meta-analysis. Fertil Steril. 2017; 108:117-124.e5.) The small number of reported cases of complications could be due to publication bias.

5. Were the IVF cycles consecutive or for how long were the patients included in the study?

Response: The IVF cycles were consecutive, and patients were included from May 2013 until May 2017. We have identified all the IVF cycles following sclerotherapy up to the point of a live birth, or until May 2020.

6. In the EST group if there were 4 pregnancy lost, the live birth rate should be 20…or maybe there are some patients in both groups…can you explain it?

Response: This is clearly a wrong categorization of pregnancies on the first analysis (biochemical pregnancies were not counted as miscarriages and an extra uterine pregnancy was not counted as a clinical pregnancy). The new clinical pregnancy rates, biochemical pregnancy rates and pregnancy loss have been corrected in Table 1 and in the abstract and corpus text as follows:

Line 17: “The clinical and biochemical pregnancy rates were significantly increased in the EST group (35.8 37.3% vs. 15.9%, p = 0.02 0.01 and 43.3% vs. 23.2%, p = 0.01, respectively).”

Line 201: “Clinical and biochemical pregnancy rates were significantly increased in the EST group (35.8 37.3% vs. 15.9%, p = 0.02 0.01 and 43.3% vs. 23.2%, p = 0.01, respectively). The differences in concerning pregnancy loss rate and implantation rate was were not significant between the two groups (respectively 13.3 26.7% vs. 6.3 37.5%, p = 0.47 0.45 and 21.0% vs. 10.7%, p = 0.06).”

Discussion

1. The weak and strong points of this work should be more highlighted in the discussion

Response: The discussion has been corrected according to your comments, as follows:

- weak points:

Line 318: “Despite bias induced by a retrospective survey, including the possibility of confounding factors, our results suggest that EST associated to with a long agonist protocol could be an interesting technique to improve IVF success rates in women with moderate-severe endometriosis. To ensure the comparability of our two groups, we performed a matching procedure based on age, BMI and smoking status, which are potential confounding factors.

We limited our analysis to small to medium endometriomas sizes (25-65 mm of diameter); therefore, our results cannot be extrapolated to larger cysts. The EST procedure was based on Yasbeck's protocol and Cohen’s meta-analysis (19,22,32), which both describe the sclerotherapy of cysts up to 65 mm. However, in our experience, we have gradually increased the size of endometriomas eligible for sclerotherapy from 65 to 100 mm in diameter. A recent study also reports the successful sclerotherapy of endometriomas up to 100 mm (51). It would be interesting to study IVF live birth rates after EST of cyst measuring 65 to 100 mm. Moreover, the management of endometriosis should be multidisciplinary, reproducible and standardized to improve the quality of care (52). The development of reference centers dedicated to the care of women with endometriosis is potentially useful to achieve this goal.”

- strong points: 

Line 299: “We were able to access the CLBR without loss to follow-up, whereas previous studies have mainly focused on ovarian response to COS and clinical pregnancy rate. Evaluating CLBR should be the main endpoint for fertility studies, especially in women with endometriosis, as they are supposedly more at an increased risk for pregnancy loss (50). Another strength of our study is the significant number of EST procedures described given EST is not a routine procedure in many centers and only small numbers are available in the literature. Moreover, we examined the medical records thoroughly and extracted a large number of variables from the fertility and surgery files, which allowed us to analyze the essential information in detail.”

Bibliography

1. Reference 31 and 40 are the same

Response: Thank you for pointing out that error. The correction has been done. 

2. More recent papers maybe useful:

a. Findeklee S, Radosa JC, Hamza A, Haj Hamoud B, Iordache I, Sklavounos P, et al. Treatment algorithm for women with endometriosis in a certified Endometriosis Unit. Minerva Ginecol 2020

b. Garcia-Tejedor A, Martinez-Garcia JM, Candas B, Suarez E, Mañalich L, Gomez M, et al. Ethanol Sclerotherapy versus Laparoscopic Surgery for Endometrioma Treatment. A Prospective, Multicenter, Cohort Pilot Study. J Minim Invasive Gynecol 2020.

Response: Thank you for these recent references. They have been added to the manuscript:

Line 328: “A recent study also reports the successful sclerotherapy of endometriomas up to 100 mm (51).”

Line 330: “Moreover, the management of endometriosis should be multidisciplinary, reproducible and standardized to improve the quality of care (52).”

Reviewer #3: The manuscript is well written and the topic is relevant.

Response: Thank you very much for your helpful comments to improve our manuscript. The responses and corrections are listed below, and changes have been highlighted in the manuscript.

Introduction: well written

We are very grateful for your sound remark.

Methods:

- did authors performed a sample size calculation or power analysis?

A post-hoc power analysis was performed, and found a power of 68% (Donner, A. and Klar, N. 2000. Design and Analysis of Cluster Randomization Trials in Health Research. Arnold. London.) The manuscript has been corrected accordingly:

Line 163: “A post-hoc power analysis was performed, and a power of 68% was identified (39)”

- was there any drop out/ lost of follow up?

Response: Thank you for pointing out that this information was missing. There was no loss to follow-up. This has been added to the manuscript, as follows: 

Line 179: “There was no loss to follow-up.”

- the results were evaluated by intention to treat? was there any cross between groups?

Response: As this is a retrospective cohort study, the “intention to treat” analysis was not applicable. Moreover, there was no possible cross between groups, as at the time of the inclusion, the patient had already undergone the ethanol sclerotherapy procedure or not before the IVF cycle.

- patients with adenomyosis were included?

Response: Yes, some of the patients included were diagnosed with adenomyosis. This result has been added to the methods and results sections (Table 1):

Line 168:

Adenomyosis 5 (13.5) 3 (8.1) 0.71

Results:

Could the authors include in table 1/results:

- the type of endometriosis and others sites affected (bowel, vaginal, retrocervical, etc)?

Response: We used Revised ENZIAN-classification for deep endometriosis in addition to the rAFS classification as recommended by Johnson et al. (Johnson NP, Hummelshoj L, Adamson GD, Keckstein J, Taylor HS, Abrao MS, et al. World Endometriosis Society consensus on the classification of endometriosis. Hum Reprod. 2017 Feb 1;32(2):315–24). This result has been added to the methods and results sections as requested in Table 1:

Line 168:

Extra ovarian endometriosis 26 (70.3) 25 (67.6) 0.80

Type of extra ovarian endometriosis 

Rectovaginal space or vaginal 11 (29.7) 15 (40.5) 0.33

Uterosacral and cardinal ligaments 21 (56.8) 15 (40.5) 0.16

Rectosigmoid 13 (35.1) 11 (29.7) 0.62

Bladder or ureter 2 (5.4) 4 (10.8) 0.67

Other bowel involvement and extragenital localizations 2 (5.4) 2 (5.4) 1.00

Adenomyosis 5 (13.5) 3 (8.1) 0.71

- number of ovaryan endometriomas, unilateral/bylateral, and mean size (before and after EST)?

Response: This result has been added to the methods and results sections as requested to Table 1: 

Line 168:

Number of endometrioma 2.1 ± 1.3 2.1 ± 1.3 0.86

Bilateral endometriomas 22 (59.5) 17 (45.9) 0.24

About the mean size after, we had it noted in the text:

Line 207: “The mean diameter of the endometriomas eligible at EST in the EST group was 37.3 ± 9.4 mm. In the No-EST group, the mean diameter of the endometriomas eligible at EST was 34.6 ± 11.4 mm. The difference was not significant between the two groups (p = 0.27)”

About the mean size before EST, we had added it in the text:

Line 210: « The mean reduction of endometriomas after EST was 28.2% ± 10.6% with a mean final diameter of 9.1 ± 9.5 mm (p <0.01).”

- previous ovaryan surgery (cystectomy, drainage, laser. ooforectomy, etc)?

Response: You can find this information on the text, on Result section. 

Line 181: “In the EST group, 43.2% of women had a history of previous laparoscopy for endometrioma treatment vs. 37.8% in the No-EST group (p = 0.64)”

- did the authors adjusted the results by age?

Response: Thank you for this remark. The results were initially not adjusted by age, because the mean age was not statistically different between women in the Ethanol sclerotherapy for endometrioma (EST) group and No-EST group (31.5 ± 4.5 years for the EST group vs. 33.0 ± 3.8 years for the No-EST group, p=0.13). However, we performed a multivariate analysis of the main predictors of live birth, including age, smoking status, and number of previous IVF cycles. After adjustment on these variables, ethanol sclerotherapy was still significantly associated with a higher chance of having a live birth. In women exposed to EST before IVF, the adjusted Odds Ratio of live birth was 2.68 (95%Confidence Interval, CI: 1.13–6.36), p = 0.02. The manuscript was corrected accordingly: 

Line 19: " Multivariate analysis revealed a significantly increased chance of live birth in women exposed to EST before IVF with an adjusted OR of 2.68 (95% confidence interval, CI: 1.13–6.36, p = 0.02)."

Line 161: “A multivariate logistic regression model was used to assess the adjusted odds ratio (aOR) of live birth. The model included classical variables known to influence the chances of success (live birth): age, smoking status, and number of previous IVF cycles.”

Line 198: " Multivariate analysis showed a significantly increased the chance of live birth in women exposed to EST before IVF. The adjusted OR of live birth was 2.68 (95% confidence interval, CI: 1.13–6.36 p = 0.02)"

- any of included patients were submmited to surgery before the IVF?

Response: We did not exclude women with a history of previous laparoscopy (that could include cystectomy), if they presented with an endometrioma recurrence at the time of the IVF cycle that was studied. 

Line 181: “In the EST group, 43.2% of women had a history of previous laparoscopy for endometrioma treatment vs. 37.8% in the No-EST group (p = 0.64).”

We thank you for pointing out that our formulation was not clear. We have added the following formulation in Table 1 accordingly:

Line 168:

History of operative laparoscopy for endometriosis global treatment 14 (37.8) 13 (35.1) 0.81

Reviewer #4: I read with great interest the Manuscript titled “Endometrioma ethanol sclerotherapy could increase IVF live birth rate in women with severe endometriosis” (PONE-D-20-17318).

The study was approved by the aix Marseille University's ethics committee (2018-24-01-008), and all women gave a written informed consent. The study was aimed to examine the impact of ethanol sclerotherapy for endometriomas (EST) on IVF cumulative live birth rates (CLBR) in women with severe endometriosis.

I was particularly pleased to review this paper. In my honest opinion, the topic is interesting enough to attract the readers’ attention. Nevertheless, authors should clarify some point and improve the discussion citing relevant and novel key articles about the topic and discussion limitations of the study that are not evidenced in the discussion:

Response: We are very grateful for your sound remarks and helpful comments to improve our manuscript. The responses and corrections are listed below, and changes have been highlighted in the manuscript.

- All the text needs a language revision by a native English speaker person, in order to improve some typos and grammatical errors.

The text has been revised by American Journal Expert, as recommended. 

- I would suggest checking the guidelines for the Authors to conform the manuscript.

Response: The guidelines for the Authors have been carefully checked, and the manuscript has been conformed accordingly as suggested.

- All the abbreviation should be reported in the extended form at the first use both in the abstract and the main text and tables.

Response: The correction has been done and changes have been highlighted in the manuscript:

Line 2:” To examine the impact of ethanol sclerotherapy for endometriomas (EST) for endometrioma on In Vitro Fertilization (IVF) cumulative live birth rates (CLBR) in women with moderate-severe endometriosis.”

Line 5: “This retrospective cohort study included women with moderate-severe endometriosis (revised American Fertility Society stage III-IV) and endometrioma(s) who underwent IVF with ultra-long agonist protocol.”

Line 80: “The diagnostic was made by trained gynecologists on transvaginal ultrasound, and then confirmed by pelvic magnetic resonance imaging (MRI) for all women by expert radiologists specialized in endometriosis imaging.”

Line 91: “Individual characteristics considered included were: age, BMI, smoking status, ovarian reserve (antral follicle count on day 3 and Anti-Müllerian Hormone (AMH) dosage), endometriosis severity evaluated by MRI or laparoscopy, duration of infertility, association of a male factor, number of previous IVF cycles, and history of laparoscopy”

Line 118: “As recommended by the European Society of Human Reproduction and Embryology (ESHRE), the IVF protocol was an ultra-long-agonist protocol consisting in of an intra-muscular injection of 3 mg Triptoreline 3 mg (Decapeptyl, Ipsen, France) at least three months before the EST procedure (11).”

Line 123: “All women underwent serial transvaginal ultrasounds and hormonal dosages during stimulation, and human chorionic gonadotropin (HCG) was administered subcutaneously (Ovitrelle, Merck Serono, Germany) when three or more leading follicles with a mean diameter greater than 18 mm were visualized. Oocyte retrieval was carried out performed transvaginally 36h later. Intracytoplasmic sperm injection (ICSI) was carried out performed if necessary, according to the sperm characteristics.”

- Abstract. Methods are lacking key information, such as the treatment investigated and how and which groups were selected and compared.

Response: We agree with that comment. The abstract has been corrected as followed:

Line 5: “This retrospective cohort study included women with moderate-severe endometriosis (revised American Fertility Society stage III-IV) and endometrioma(s) who underwent IVF with ultra-long agonist protocol. We compared two groups: women undergoing EST for endometrioma before IVF (EST group), and women whose endometrioma was left in situ during IVF (No-EST group).”

- Methods. I would suggest providing the following pieces of information:

a) How the patients were identified

b) The source of patient list (medical records, registry, DRG)

Response: The patient list came from the hospital database including all IVF and ICSI cycles performed at our center between May 2013 and May 2017. This list is exhaustive, as each IVF center in France has the obligation to report all Assisted Reproductive Technologies treatments to the national French registry of IVF. We then examined all medical records thoroughly and extracted a large number of variables from the fertility and surgery files. These variables come from a specific software used in ART called Medifirst.

c) Who extracted data?

Response: Data were extracted by two physicians of the Reproductive Medicine Unit (Dr Laura MIQUEL and Pr Jeanne PERRIN) and analyzed by a statistician (Noemie RESSEGUIER). 

d) Source of clinical information

Response: All the clinical information of all couples followed in our reproductive medicine center are recorded in Medifirst software (medical consultation, ultrasound examination, IVF monitoring, oocyte pickup, embryo transfer etc…). 

- Regarding endometriosis score, were all patients reassessed by RMI before EST/IVF regardless surgery or for some patients was used the score estimated during surgery? In this case the effect of surgery represents a factor to be considered in the comparison of the two groups. In this regard, is the RMI accuracy valid for patients who had already underwent surgery?

Response: As you pointed out, all women were reassessed by RMI before EST/IVF regardless surgery.

- The match of patients should be reported in the statistical analysis description.

Thank you for this remark. The matching scheme has been added to the statistical analysis description, as follows: 

Line 150: “Women exposed to EST were selected and matched in a 1:1 fashion to women who were not exposed to EST. Matching criteria included age (+/- 2 years), BMI (+/- 3 kgs/m2), and smoking status.”

- Statistical analysis. Was the assumption of normal distribution for continues variable assessed?

Response: Assumption of normality of the distribution for continuous variables was assessed graphically (histograms and quantile-quantile plots). Median and [1st quartile- 3rd quartile] were presented when assumption of normality was not verified. This statement was added to the statistical analysis section:

Line 158: “Assumption of normality of the distribution for continuous variables was assessed graphically (histograms and quantile-quantile plots). Median and [1st quartile- 3rd quartile] were presented when assumption of normality was not verified.” 

- Results. The matching of the two groups needs to be better described. 

How many patients with severe endometriosis underwent EST? How many were eligible for the study? How many patients did not undergo EST? 

Response: A total of 37 women with severe endometriosis underwent EST during the study period, and were all eligible for the study. We thank you for pointing out that our formulation was not clear. We have added the following sentences accordingly:

Line 167: “Among those 294 women, 191 women had ovarian endometriosis. Among those 191 women, 158 women still had ovarian endometriosis immediately before IVF. Among these 158 women, 123 women had endometrioma measuring 25 to 65 mm in diameter. Among these 123 women, 6 women underwent a laparoscopic management of the endometrioma immediately before IVF, 77 women did not undergo any treatment and 40 woman underwent sclerotherapy. Among these 117 woman eligible for the study, 99 give their consent the study. Among those 99 woman selected for the study, 37 women had undergone EST and were matched in a 1:1 fashion to women who were not exposed to EST.”

How the two groups were matched based on age, BMI, and smoke? In the example, were matched by age based on age +- 1 year? 

Response : As previously pointed out thanks to your remark, the matching scheme has been added to the statistical analysis description, as follows: 

Line 150: “Women exposed to EST were selected and matched in a 1:1 fashion to women who were not exposed to EST. Matching criteria included age (+/- 2 years), BMI (+/- 3 kgs/m2), and smoking status.”

Being possible to match 1:2 or 1:3, did the authors estimated a sample size calculation? 

Response: Owing to the small number of eligible patients for the study, a 1:2 or 1:3 matching was not possible:

Line 173: “Among these 117 woman eligible for the study, 99 give their consent the study. Among those 99 woman selected for the study, 37 women had undergone EST and were matched in a 1:1 fashion to women who were not exposed to EST.”

We initially did not estimate a sample size calculation. However, a post-hoc power analysis was performed, and found a power of 68% (Donner, A. and Klar, N. 2000. Design and Analysis of Cluster Randomization Trials in Health Research. Arnold. London.) The manuscript has been corrected accordingly:

Line 163: “A post-hoc power analysis was performed, and a power of 68% was identified (39)”

Were matched women who did not received EST randomly selected? Or was a propensity score method used?

Response: We did not use a propensity score method for that selection because we have selected all the patients in our center between May 2013 and May 2017:

Line 166: “Among 2148 women who underwent IVF-ICSI in our center between May 2013 and May 2017, 381 had endometriosis, and 294 had moderate-severe endometriosis. Among those 294 women, 191 women had ovarian endometriosis. Among those 191 women, 158 women still had ovarian endometriosis immediately before IVF. Among these 158 women, 123 women had endometrioma measuring 25 to 65 mm in diameter. Among these 123 women, 6 women underwent a laparoscopic management of the endometrioma immediately before IVF, 77 women did not undergo any treatment and 40 woman underwent sclerotherapy. Among these 117 woman eligible for the study, 99 give their consent the study. Among those 99 woman selected for the study, 37 women had undergone EST and were matched in a 1:1 fashion to women who were not exposed to EST.”

- Media follow-up among patients who obtained a pregnancy should be reported. How many patients were missed during follow-up? At which time point does the CLBR refer to?

Response: Thank you for pointing out that this information was missing. There was no loss to follow-up. This has been added to the manuscript, as follows: 

Line 179: “There was no loss to follow-up.”

About the time point of the CLBR, if a women achieved pregnancy, we stopped the followed up. If they does not, we contact all the women on May 2020 :

Line 74: “All women were followed-up until May 2020.”

Line 136: “Only the first delivery was counted in the analysis if a patient achieved multiple deliveries.”

We have added the following sentences accordingly:

Line 175: “The median follow-up among patients who obtained a pregnancy was 13.9 ± 7.7 months for the EST group and 9.2 ± 8.5 months for the No EST group. The median follow-up among patients who did not obtained a pregnancy was 51.1 ± 10.1 months for the EST group and 72.0 ± 18.9 months for the No EST group.”

- I would suggest improving the discussion discussing at least briefly about the etiopathogenesis of endometriosis. Refer to (PMID: 32046116; PMID: 31717614).

Response: Thank you for your comment and these references. The discussion has been modified accordingly, as follows: 

Line 255: “Endometriosis is a chronic, inflammatory, estrogen-dependent disease, in which endometrial stromal cells acquire the capacity to proliferate, migrate outside the uterine cavity, and invade adjacent tissues. One of the main hypotheses is based on the retrograde menstruation (34). However, the mechanisms underlying the endometrial tissue grafting essentially remain unknown. Recent studies support a multifactorial origin, combining anatomical, hormonal, immunological, genetic, epigenetic, and environmental factors (45).”

Line 261: “Considering the molecular level, it appears that ion channels, such as cystic fibrosis transmembrane conductance regulator, aquaporins, and chloride channel-3 dysregulation are potentially involved in the physiopathology of endometriosis (46).”

- I would suggest, at least briefly, to discuss more about the COS in patients with endometriosis and specifically endometriomas (PMID: 31755673).

Response: The discussion has been modified as follows:

Line 235: “Compared to controls, women with endometriomas have a lower ovarian reserve and ovarian response to COS as demonstrated by the reduced number of retrieved oocytes and mature oocytes (40,41). The impact of IVF on an endometriotic cyst seems to be mild. In their systematic review, Somigliana and al. reported that IVF does not worsen endometriosis-related pain symptoms or increase the risk of endometriosis recurrence (42).”

---

## [Decision Letter · Decision Letter 1]

15 Sep 2020

Endometrioma ethanol sclerotherapy could increase IVF live birth rate in women with moderate-severe endometriosis

PONE-D-20-17318R1

Dear Dr. MIQUEL,

We’re pleased to inform you that your manuscript has been judged scientifically suitable for publication and will be formally accepted for publication once it meets all outstanding technical requirements.

Kind regards,

Simone Garzon

Academic Editor

PLOS ONE

Additional Editor Comments (optional):

Reviewers' comments:

Reviewer's Responses to Questions

**Comments to the Author**

1. If the authors have adequately addressed your comments raised in a previous round of review and you feel that this manuscript is now acceptable for publication, you may indicate that here to bypass the “Comments to the Author” section, enter your conflict of interest statement in the “Confidential to Editor” section, and submit your "Accept" recommendation.

Reviewer #1: (No Response)

Reviewer #4: All comments have been addressed

2. Is the manuscript technically sound, and do the data support the conclusions?

Reviewer #1: (No Response)

Reviewer #4: Yes

3. Has the statistical analysis been performed appropriately and rigorously? 

Reviewer #1: (No Response)

Reviewer #4: Yes

4. Have the authors made all data underlying the findings in their manuscript fully available?

Reviewer #1: (No Response)

Reviewer #4: Yes

5. Is the manuscript presented in an intelligible fashion and written in standard English?

Reviewer #1: (No Response)

Reviewer #4: Yes

6. Review Comments to the Author

Reviewer #1: (No Response)

Reviewer #4: I carefully evaluated the revised version of this manuscript.

Authors have performed the required changes, improving significantly the quality of the paper.

7. PLOS authors have the option to publish the peer review history of their article (what does this mean?). If published, this will include your full peer review and any attached files.

Reviewer #1: No

Reviewer #4: No

---

## [Editor Report · Acceptance letter]

17 Sep 2020

PONE-D-20-17318R1 

Endometrioma ethanol sclerotherapy could increase IVF live birth rate in women with moderate-severe endometriosis 

Dear Dr. Miquel:

I'm pleased to inform you that your manuscript has been deemed suitable for publication in PLOS ONE. Congratulations! Your manuscript is now with our production department. 

Kind regards, 

on behalf of

Dr. Simone Garzon 

Academic Editor

PLOS ONE